# Profiles of Monocyte Subsets and Fibrosis-Related Genes in Patients with Muscular Dystrophy Undergoing Intermittent Prednisone Therapy

**DOI:** 10.3390/ijms26135992

**Published:** 2025-06-22

**Authors:** Asma Chikhaoui, Dorra Najjar, Sami Bouchoucha, Rim Boussetta, Nadia Ben Achour, Kalthoum Tizaoui, Ichraf Kraoua, Ilhem Turki, Houda Yacoub-Youssef

**Affiliations:** 1Laboratory of Biomedical Genomics and Oncogenetics (LR16IPT05), Institut Pasteur de Tunis, Université Tunis El Manar, El Manar I, Tunis 2092, Tunisia; 2Service Orthopédie, Hôpital d’enfant Béchir Hamza, Tunis 1007, Tunisia; 3LR18SP04, Department of Child and Adolescent Neurology, National Institute Mongi Ben Hmida of Neurology, Tunis 1007, Tunisia

**Keywords:** monocyte phenotypes, gene expression, muscle dystrophies, prednisone

## Abstract

Muscle dystrophies are a group of genetic disorders characterized by progressive muscle degeneration. Prednisone is a glucocorticoid drug widely used to prevent muscle weakness in these diseases. Despite its known beneficial role, the effect of intermittent delivery on monocytes’ polarization and on dystrophic muscle microenvironment has not yet been thoroughly investigated. In this study, our aim was to identify the phenotype of monocyte subsets in blood and the expression of fibrosis-related genes in dystrophic muscle biopsies in patients receiving intermittent prednisone therapy. We found an increased rate of classical monocytes and a decreased rate of non-classical monocytes that expressed anti-inflammatory marker CD206 in treated patients. In dystrophic muscles, 21 fibrosis-related genes were altered, among which we identified CCAAT/enhancer-binding protein beta *CEBPB*. Both classical monocytes and *CEBPB* are known for their roles in stimulating collagen 1 production, a probable marker hampering monocyte/macrophage function. Hence, in some patients with muscular dystrophy, intermittent prednisone treatment could shift the monocytes’ phenotype toward an M2, senescent-like profile. This seems to decrease the inflammatory infiltrate in muscle tissue, an observation that needs to be further confirmed.

## 1. Introduction

Muscular dystrophies (MDs) are a group of degenerative and hereditary disorders that cause progressive skeletal muscle loss. They represent more than thirty subtypes, with the most prevalent ones in Tunisia being limb–girdle muscular dystrophy (LGMD) (0.6/2000) [1] and Duchenne muscular dystrophy (DMD), having a prevalence of 7 × 10^−5^, with both caused by genes encoding the proteins of the dystrophin-associated glycoprotein complex [2,3]. DMD is caused by mutations in the X chromosome coding in the dystrophin gene, causing progressive muscle degeneration, severe disability, and premature death due to cardiac failure. It affects boys with a prevalence of 1 per 5000 live births worldwide [4,5].

LGMD is a hereditary muscular disorder characterized by muscle weakness in the shoulder and pelvic girdles, mainly due to mutations in Sarcoglycan genes. There are several subtypes of LGMD; however, in our study, we were interested in the LGMD2C (LGMD/R5) form, which is linked to mutations in the SGCG gene [6].

Muscles in MDs possess a limited satellite cell recovery rate, which contributes to their degenerative characteristics. Dystrophic muscles in both DMD and LGMDR5 contain a significant alteration in the functions of stromal cells, particularly macrophages, due to the particular microenvironment in which they are located, which has an impact on their functioning [7,8].

Prednisone (Pred) is the most frequently used treatment for MDs. It is a glucocorticoid (GC), a potent anti-inflammatory and immunosuppressant drug that affects cells through genomic and non-genomic mechanisms [9,10]. Pred is used to treat patients with MDs by decreasing immune cell infiltration within the muscles, which is one of the main features of muscular dystrophy diseases [11]. However, the advantage of using a GC in treating MD muscles is most likely due to a combination of processes that balance the positive and negative effects, alongside possible cases of glucocorticoid resistance [12]. Many clinical studies have shown that corticosteroids can delay MDs from worsening; however, their effectiveness remains unclear because of their severe and transitory side effects [13].

In the case of LGMD patients, specific therapeutic interventions are available for the management of particular symptoms, such as the use of nocturnal ventilation for respiratory dysfunction and β-blockers against cardiac manifestations [14,15]. Interestingly, only patients with the LGMD/R5 form showed improvement with prednisone administration, which has been employed in North African patients since the 1990s [16,17,18]. Pred dosing in MDs varies according to treatment strategies, spanning daily doses, high weekly doses, and ten-day on/off cycles [19]. Previous work on mouse models of LGMD and mdx mice suggested a reduction in the main inflammatory responses with once-weekly doses depending on the time of administration [20]. However, other studies that used animal models suggested that the long-term evaluation of GC could not be adequately assessed as fibrosis and muscle deterioration occur [21].

Data regarding administration strategies in patients differ: Some data suggest that prednisone treatment according to the 10 on/10 off protocol has fewer side effects and extends the ambulant phase by 1 year in DMD patients [22,23], while other data suggest the opposite [19,24]. In addition, it has become more evident that evaluating the dosing regimen’s success in mice does not always guarantee the same results in humans [25]. In light of these observations, it has become more evident that the choice of corticosteroid depends on the clinician’s decision [26]. While previous investigations have been conducted on boys with Duchenne muscular dystrophy [19], there are few randomized datasets exploring other muscular myopathies.

Both innate and adaptive immune responses are involved in the regeneration of skeletal muscle and muscular dystrophies [27]. Following signals from damaged muscle, the innate immune response is activated by myogenic precursor cells, which recruit monocytes to the site of injury [28,29]. Different subsets of monocytes have been identified according to the expression of surface receptors CD14 and CD16. Classical CD14++CD16−, intermediate CD14++CD16+, and non-classical CD14+CD16++ monocytes are the main subsets characterized by most researchers [30,31]. Recruited monocytes in damaged muscle generally differentiate into macrophages, which may be polarized into pro-inflammatory M1 macrophages during the early stage following injury or switch into anti-inflammatory phenotype M2 macrophages that contribute to the later stage of tissue repair [32,33]. These two phenotypes play a dual role, firstly by stimulating myocyte proliferation and then, in a second phase, supporting the differentiation of myoblasts into myotubes [34]. In mouse models, monocytes have been identified as the most influential players in muscle fibrosis [35]. While changes in human monocyte counts and subsets have been observed in several fibrotic disorders [36], less is known about those in MDs [37]. Recent studies suggest that resident macrophages play a role in the metabolic reprogramming of skeletal muscle fiber compositions, suggesting a different role for macrophage-derived monocytes [38].

Due to continuous cycles of the degeneration/regeneration of muscle fibers in MDs, M1 and M2 macrophages are constantly present at the site of muscle injury [39,40]. In particular, pro-inflammatory macrophages in DMD contribute to fibrosis by producing different secretory factors [41], highlighting their potential role in fibrogenesis.

In this preliminary observational study, we investigated the effect of intermittent Pred treatment on a limited set of phenotypes from patient monocytes, as these are the main actors of inflammation and play a decisive role in muscle regeneration. We also analyzed the expression of a set of genes involved in the fibrosis process, which is the determining factor causing muscle degeneration in MD patients.

## 2. Results

### 2.1. Intermittent Prednisone Induces an Alterations in Monocyte Subsets in Muscle Dystrophies

To investigate the effect of 10-day on/off Pred administration on the monocytes of patients with muscle dystrophies, we collected peripheral blood mononuclear cells from nine patients. Among them, five were assigned to the “steroid-free group” (MD group) since they did not receive any treatment at the time of this study, and four received intermittent treatment during the study (MD-Pred). Flow cytometry analysis showed that the number of total monocytes was higher in the blood of MD-Pred. In detail, the rate of monocytes was low in the MD group at 1.550 ± 0.6990 compared to healthy controls (HC, N = 15): 2.839 ± 1.628. With intermittent Pred treatment, the percentage of total monocytes increased significantly to 5.080 ± 1.677; *p*-value = 0.0051 (Figure 1b). We also tried to characterize all three monocyte subsets: classical monocytes (CD14++CD16-), intermediate monocytes (CD14++CD16+), and non-classical monocytes (CD14+/CD16++). Among the different subsets, MD-Pred patients had a higher classical monocyte rate of 91.63 ± 2.109 compared to HC (80.12 ± 7.019: *p*-value = 0.0147). Lower rates of non-classical monocytes were also observed after Pred treatment (MD-Pred 4.190 ± 4, HC 15.74 ± 7.552; *p*-value = 0.0282) (Figure 1c).

### 2.2. Expression of Surface Receptors CD86, CD206, and CD163 in Monocyte Subsets Shifts Toward an M2-like Phenotype After Intermittent Prednisone Treatment

Among the assessed monocyte subsets, there were variations in the expression patterns of CD86, CD206, and CD163. The MFI (mean fluorescence intensity) of CD86 increased in classical and intermediate monocytes in the MD group. After Pred treatment, CD86 expression decreased in classical monocytes with a fold change of 0.359 ± 0.3; *p*-value = 0.0052 (Figure 2a).

The expression of CD163, the macrophage scavenger receptor, was higher in all subsets of monocytes in MD-Pred patients. Furthermore, the MFI of CD163, assessed by flow cytometry, was higher in MD-Pred patients’ intermediate monocytes compared to healthy controls (HC 168.9 ± 153; MD-Pred 394.3 ± 303, *p*-value = 0.0177). This was also confirmed by QPCR analysis compared to the healthy controls (fold change MD-Pred = 88.8 ± 210, *p*-value < 0.001) (Figure 2b).

Although there were differences in CD206 expression between the monocyte subsets, it was generally similar to the expression of CD163, with a predominance of classical and intermediate monocytes. The level of CD206 MFI was significantly lower in MD patients’ intermediate monocytes compared to healthy controls (HC = 1157.2 ± 469, MD = 671.85 ± 473, *p*-value = 0.04). In the MD-Pred group, the CD206 level is increased in the non-classical monocyte subset compared to healthy controls (fold change MD-pred 269 ± 100, *p*-value = 0.01) (Figure 2c).

### 2.3. Assessment of Intermittent Prednisone Treatment Effects on Muscle Biopsies in Patients with Muscular Dystrophy

Hematoxylin–eosin was used to assess inflammatory cell infiltration, and Masson’s trichrome staining was used to assess fibrosis in muscle biopsies from MD-Pred and MD patients. Although Masson’s trichrome staining of collagen deposits appeared similar in both groups (average percentage of 30% in both groups), we observed that the rate of inflammatory cells was lower in the MD-Pred group (score 1) than in the MD group (score of 2) (Figure 3).

### 2.4. Intermittent Prednisone Exposure Induces Dynamic Changes in Gene Expression in Muscle Biopsies

By pathway-focused PCR array analysis, we comprehensively investigated the expression of genes involved in fibrogenesis in three MD patients and three MD-Pred patients. After the 10-day on/off regime, the expression of 21 genes was altered. We created a heat map to show gene expression profiles (Figure 4a) and further explored their involvement through an in silico study of related protein–protein interactions (PPIs). Through the PPIs, the STRING tool demonstrated three main clusters of gene interactions (pathways) (Figure 4b). The IL4-IL13 signaling pathway seemed to be a common pathway, relating 8 of the total 21 protein interactors (MMP2, MMP1, FASLG, CCL11, HGF, STAT6, IL13, and IL4), which was also shown in the Reactome database with a *p*-value score of 4.44 × 10^−16^.

Given the involvement of the IL4/IL13 pathway in immune responses, we further explored the expression of altered genes in the public database “immgen” of Single-Cell RNA-seq profiles (Figure 4c). The corresponding dot plot for the available genes showed that the highest expression of *CEBPB* was observed in classical monocytes at 43%, and in non-classical monocytes, the highest expression was 76% compared to other immune cell populations, as shown in the Single Cell RNA-seq profiles public data and compared to the rest of the altered gene set.

## 3. Discussion

### 3.1. Overview of Prednisone Regimens in Muscle Dystrophies

Glucocorticoids (GCs) are commonly prescribed immunosuppressive therapies for chronic inflammatory diseases. Prednisone (Pred) is a synthetic glucocorticoid that is metabolized in the liver to its active form, prednisolone [42,43]. Previous studies confirm its beneficial roles in muscle dystrophies, resulting in a prolongation of walking by ~2 years [44]. However, despite their use as the therapy of choice in muscular dystrophies, there are substantial side effects, such as diabetes and osteoporosis due to weight gain, followed by spontaneous fractures and cataracts [45,46,47].

This is why intermittent dosing has been suggested as a possible alternative to these effects; however, it is not clear exactly how long the effect lasts or how it affects various components of the muscle microenvironment, including the immune players that are ubiquitous and play a crucial role in the pathophysiology of MDs [48]. Previous studies using the 10-day on/off administration showed better outcomes for the daily regimen compared to the intermittent one [19]. Furthermore, intermittent prednisone dosing was associated with the frequent emergence of some side effects, such as cushingoid appearance, excessive hair growth, and hypertension [24]. However, little is known regarding the exact impact on immune cells and muscle tissue [12].

### 3.2. Intermittent Prednisone Induces Changes in Monocyte Subset Phenotypes

Hematopoietic stem cells in the bone marrow give rise to monocytes, which are then discharged into peripheral tissues through the bloodstream [49]. Following intermittent Pred treatment, we observed an increase in monocytes in MD patients; this observation was similar to what was noted after glucocorticoid treatment in other disease models, such the osteonecrosis of the femoral head [50]. This may be related to the increased survival of monocytes following treatment [51,52].

In humans, monocytes play different roles. Classical monocytes are the first group that infiltrate the tissue, playing a crucial role in phagocytosis and the production of pro-apoptotic mediators [53]. The pool of classical monocytes could give rise to intermediate and non-classical monocyte subsets [54]. The intermediate population plays a role in angiogenesis and antigen presentation [55], while non-classical monocytes participate in the removal of damaged cells and the maintenance of vascular homeostasis [56]. In MDs, the role of macrophages in muscle regeneration is widely described [57,58]. However, less is known about any phenotypic changes in monocytes before they differentiate into macrophages within the muscle niche.

In this work, we found that the rate of classical monocytes increased, while the rate of non-classical monocytes decreased in Pred-MD patients compared with untreated MD patients and with the healthy donor group. Previous work showed that classical monocytes play a key role in the excessive formation of osteoclasts and contribute to GC-induced bone loss [50]. As in patients with muscle dystrophies, such as in DMD, there is an increased risk of bone fragility due to the adverse effects of prolonged glucocorticoid therapy [59]. The clinical relevance of our findings was also demonstrated in other disorders due to the Pred treatment, such as idiopathic inflammatory myopathies, and in acute renal allograft rejection [60,61]. Monocytes, being the main precursors of active macrophages within the muscle, were examined in depth in this study, particularly in terms of M1- and M2-like profiles.

The analysis of the effect of the 10-day on/off Pred treatment in MD patients suggested that GC treatment decreased CD86 expression in monocytes, particularly in the classical subtype, which could induce anti-inflammatory effects. CD86 is known to act as a co-stimulatory molecule during T cell activation and is wildly expressed by antigen-presenting cells, particularly M1 pro-inflammatory macrophages [62]. The particularities of classical monocytes being the most affected by Pred treatment were observed in patients with thrombocytopenia [63], and they may therefore be linked to their role in muscle injury, inflammation, and regeneration, as it was stated in mdx mouse models [35]. It is also well known that Pred skews monocytes towards an anti-inflammatory model, increasing the expression level of the CD163 and CD206 genes [64]. In DMD patients, it has been reported that CD163 was also upregulated in the blood of steroid-treated patients, and it was suggested that the overexpression of haptoglobin may bind and sequester heme proteins released from damaged or dying muscle fibers in a beneficial way for muscle regeneration [65]. In our study, we observed that following MD patients’ treatment with Pred, there is an increase in intermediate monocyte subsets expressing CD163 and CD206, as has been shown in other steroid-treated disorders, and this is related to the attenuation of T cell responses [63,66]. However, the increased expression of CD163 observed in both MD and MD-Pred groups indicates that this is a characteristic feature of the MD phenotype rather than a consequence of Prednisone treatment.

Regarding CD206, it corresponds to a mannose receptor presented as a transmembrane protein that plays a crucial role in antigen uptake and presentation due to its high affinity for glycosylated antigens [67]. In our study, we found a difference between the results of the cell surface CD206 receptor’s expression, measured by MFI/cytometry, which was decreased, and the rate of the CD206 gene, measured in sorted monocytes by rt-qPCR, which increased. This discordance may be related to the presence of the soluble form of CD206 in the blood, which has been associated with many other inflammatory disorders, such as obesity-associated metainflammation [68,69]. In our case, we suggest that the intermittent Pred treatment increased the level of CD206 expression in the serum and not on the monocyte cells’ surface. At the same time, previous studies performed in mouse models showed that non-classical monocytes infiltrate skeletal muscles after volumetric injury, which is associated with a high level of CD206 that could therefore play a role in muscle repair [70]. In our study, Pred treatment increased CD206 expression, particularly in non-classical monocytes, which could have a beneficial but limited impact given the low rate of this population in the blood. However, further investigation is needed to confirm this observation.

It is important to note that the variability of the expression of CD163 and CD206 between flow cytometry and qPCR analysis may be due to different factors, such as epitope accessibility, which has been suggested to induce discrepancies in the reported levels of monocytic CD163 expression via flow cytometry [71], and the recycling between the plasma membrane and intracellular compartments of CD206 [72]. Furthermore, after activation, both markers are shed from the macrophage’s surface, generating soluble products that increase in several disease models [73,74] and for which is not known during MD. The shedding into the soluble form of both receptors may also occur without a parallel increase in the mRNA due to the post-transcriptional effect of the systemic glucocorticoids stimuli [74]. Therefore, based on the available data, we cannot make a conclusive observation, as the uncoupling between mRNA levels and surface proteins of CD206 and CD163 is not unexpected. Our first observation could suggest a possible anti-inflammatory M2-like reprogramming of muscle-associated monocytes.

### 3.3. Intermittent Prednisone Does Not Reduce Fibrosis in MD Muscles in Some Cases

Fibrosis results from a reparative process involving different mechanical, humoral, and cellular factors. It forms a mechanical barrier, leading to incomplete skeletal muscle healing [75]. In biopsy-analyzed samples of MD-Pred patients, no clear pattern was observed under these specific experimental conditions regarding the fibrosis level in the muscle niche after treatment compared to non-treated patients. These results differ from previous studies, which state that steroid treatment can reduce the amount of connective tissue [76]. The disparity could be explained by differences in age at the time of disease diagnosis (which was approximately 9 years in our case), while in the stated study, it was around 6 to 7 years. This could also be explained by the period of treatment received before performing the biopsy, which was longer in our group, suggesting a possible decrease in the effect of treatments in some cases. Further quantitative histological methods and larger sample cohorts are needed to explore these preliminary findings.

Previous studies exploring the effect of Pred supplementation in MD patients highlighted changes in gene expression in blood cells [65]. Similarly, mdx mouse models showed the upregulation of several genes related to metabolism, the modulation of proteolysis-related transcripts, and calcium ion influxes following Pred treatment [77]. In our study, the effect of Pred treatment in MD patients was examined using a PCR array, by which we analyzed 84 genes related to fibrosis. We found that 21 genes were upregulated after Pred administration compared to the untreated group. Among these genes, some were known for their anti-fibrotic role, such as BMP7 [78], while others exert the opposite role, such as MMP2 [37].

To explain the balance between these paradoxical effects, we proceed to further clustering using in silico analysis via the PPI network, which constructs three interconnected gene networks. The first network revealed high implications of the IL4/IL13 signaling pathway after intermittent Pred treatment, while the second and third networks comprised pathways related to matrix degradation and TGF-β/Smad pathways.

Only the IL4/IL13 pathway was noteworthy according to the REACTOME database, where previous work stated that a local increase in IL-4 and IL-13 stimulates fibro/adipogenic progenitors to proliferate and support muscle tissue regeneration by phagocytizing debris and producing new matrix components. This type 2 immune response also induces an anti-inflammatory response by stimulating M2 macrophage polarization [57,79]. In accordance with this, in mouse models, it has been shown that during the transition from the acute peak of muscle degeneration in the mdx model, the expression of IL-4 increases, which consequently promotes the activation of M2 cells expressing CD163+, increasing tissue repair [80]. From the listed potential protein–protein interactions of IL-13/IL-4, it is interesting to note that actors such as *STAT6* do not affect the rate or severity of fibrotic scar formation or disease progression in mdx mice [81]. Therefore, its altered expression in our findings may be more likely due to intermittent Pred effects. While most studies reported that corticosteroid, in general, inhibits IL-4 signaling by downregulating the IL-4 receptor and STAT6 activity [82], others suggest an increase in some related proteins, such as IL-13, in the DMD group treated with steroids [83]. This discordance highlights the need for a larger cohort study to counteract the possible fluctuation in the expression of biomarkers after GC according to the duration of treatment, the posology, the clinical status of the patients, and other parameters that need to be taken into account.

### 3.4. Prednisone Affects Fibrosis-Related Gene Expression: The Interesting Role of CEBPB

We studied the expression of candidate genes involved in fibrosis using the PCR-array technique. From this, we noted an increase in the expression of the gene coding for CCAAT/enhancer binding protein beta (*CEBPB*) in muscle biopsies from patients receiving intermittent treatments of Pred. This gene was further explored using RNA-seq data and seems to be highly expressed in the monocytes from healthy donors. The C/EBP proteins are a family of basic-region leucine zipper (bZIP) transcription factors that include six members, such as C/EBPβ. The latter is expressed in all cell subtypes and tissues, including muscle stem cells and immune cells. C/EBPβ is a transcription factor involved in many cell functionalities, including inflammation, and it plays a critical role in the induction and maintenance of the senescent state [84,85]. In addition, C/EBPβ is involved in monocyte differentiation and activation [86]. It was reported that the level of C/EBPβ is increased in exhausted monocytes [87], while others suggest that this gene induces M2 macrophage-specific gene expression [88]. In healthy skeletal muscles, C/EBPβ expression is the highest in satellite stem cells [89]. Several studies have shown that muscle atrophy may be mediated by glucocorticoid side effects and is associated with the increased expression and activity of transcription factor C/EBPβ [90].

Using the current available data, we suggest that C/EBPβ may act on the muscle microenvironment after Pred treatment. It increases through different factors contributing to immuno-senescence and muscle atrophy. The in silico exploration of its level of expression in the context of aging myotubes showed that this increase is mild according to the aging atlas (log2 FC = 0.4). Recently, an in-depth exploration of RNAseq data from skeletal muscles suggested that *CEBPB* could serve as an adipogenesis-related biomarker in muscle aging [91]. This could be one of the main side effects of intermittent GC treatment and the reason for muscle deterioration in the studied group, which requires further validation. An investigation of senescent-associated secretory biomarkers upon prolonged intermittent Pred treatment could help in obtaining a better understanding.

It is important to note that the small sample size of the studied cohort and the non-randomized observational design, which could result in false positives, correspond to one of the major concerns of this study. Additionally, clinical severity and disease progression can be quite variable in patients with LGMD [92] and DMD [93]. Furthermore, it is important to highlight that the observed variability in response to treatment could also be related to the difference in sex ratios within the studied cohort. In fact, it has been recently reported that intermittent glucocorticoid treatment enhances skeletal muscle performance through sexually dimorphic mechanisms [94].

It is also important to highlight the variability of gene expression analysis in response to intermittent prednisone treatment, as previously suggested [12]. The rarity of muscle biopsy samples used in this study was mainly due to the difficulties in convincing the legal guardians of patients to participate in a research study and/or the handling of surgical waste, significantly restricting investigations using immunohistological analysis, which requires a special handling process [95]. In addition, the lack of randomization was due to the coincidence that all patients who participated in this study received intermittent Prednisone treatment due to their clinician’s decision, as they seemed to have poor tolerance relative to ongoing prednisone treatment (personal communications). For these reasons, we insist that the current data represent a preliminary observational study that needs further investigation. Most of the current work represents observations, and the association is based on in silico network analysis and the existing literature and should be interpreted in future studies for further validation. We regret that functional assays and the use of animal models were beyond our current capacity due to budgetary constraints and the limited availability of patient-derived materials, as previously stated in other studies [96].

However, we emphasize that this study provides interesting preliminary data that include patients with limb–girdle muscular dystrophy LGMD2C/R5, a subgroup of muscular dystrophies that remains underrepresented in the literature and is limited to clinical or genetic description within a limited set of patients [97].

Regarding the merging of DMD and LGMD patients in the studied cohort, this was carried out by taking into account the fact that North African patients with LGMD2C/R5 mutations possess clinical features indistinguishable from DMD [2]. In addition, in our cohort, the effect of intermittent Pred was visible two years following the treatment [98], while in other populations, an improvement in muscle performance was observed 6 months later [99]. This reflects the particularity of the studied population.

In summary, the characterization of monocyte subsets in the blood, as they are known to dictate the fate of tissue macrophages after intermittent Pred treatment in MD, suggests an increase in classical monocytes, which partly explains the fibrogenesis process, as demonstrated in vitro in previous studies. This also suggests a shift toward an anti-inflammatory phenotype through the expression of CD206 and CD163. Our findings also illustrate the challenges in determining the therapeutic effects of intermittent Pred administration on dynamic tissue, such as dystrophic muscles. While a high level of non-classical monocytes expressing CD206 could help in inflammation resolution, the significant decrease in the total number of non-classical monocytes could hinder the desired efficiency of the Pred treatment. In addition, even with the potential anti-inflammatory effect of intermittent Pred treatment, the inhibition or even the decrease in fibrosis cannot be histologically assessed, and muscle weakness continues to progress in the studied cases. Our findings provide a preliminary identification of a relevant molecular target, the CCAAT/enhancer binding protein beta, that needs to be further examined to confirm the results, and novel therapeutic strategies should be explored.

## 4. Materials and Methods

### 4.1. Clinical and Genetic Data of Participants

This study was carried out in accordance with the Helsinki principles and approved by the Institute Pasteur Ethics Committee in Tunisia (reference 2015/28/I/LR11IPT05/V2).

Three groups of subjects were included in this study. The first group included patients suffering from muscle dystrophies who did not take any treatment at the moment of the study (MD). The second group received an intermittent regimen of prednisone, prescribed at a daily dose of 0.75 mg/kg for 10 days on and 10 days off, for more than one year (MD-Pred). Both groups had at least a creatine kinase level 30 times higher than normal levels. Both groups had their treatment regimen independently of this study and in accordance with their clinician’s decision. The last group corresponds to healthy age-matched controls (HCs) who did not suffer from any inflammatory condition nor take any medication at the time of this study. Additional information concerning the genetic and demographical details of MD and MD Pred patients is detailed in Table 1. The genetic/clinical phenotype was validated before analysis.

### 4.2. Preparation of PBMC

Venous blood was collected with 10 U/mL heparin, and peripheral blood mononuclear cells (PBMCs) were separated using a Ficoll gradient. The aliquots of PBMCs were stored in liquid nitrogen for cell sorting or directly used for flow cytometry analysis.

### 4.3. Monocyte Subset Flow Cytometry Analysis

Peripheral blood mononuclear cells were labeled with a range of fluorescent conjugated antibodies for monocyte/macrophage characterization: anti-CD14-APC, anti-CD16-FITC, anti-CD206-PE cy5, anti-CD163-PE A, and anti-CD86-PE cy7 (BD bioscience, Franklin Lakes, NJ, USA). Incubation was carried out for 30 min at 4 °C in the dark. Event acquisition was carried out using the BD FACS Canto II flow cytometer and analyzed with Flowjo software (version 10.0).

### 4.4. FACS Sorting of Monocyte

After thawing and the suspension of frozen PBMC, monocytes were selected based on FSC vs. SSC and CD14 expression. The three monocyte subpopulations were then selected according to CD14 and CD16 expression intensities: CD14++CD16- for classical monocytes, CD14++CD16+ for intermediate monocytes, and CD14+CD16++ for non-classical monocytes. Sorting was carried out on a MoFLOAstrios “Beckman Coulter” at the CRT Technology Cores of Pasteur Institute, Paris. The low cell counts expected after resuspension led us to place the sorted pool directly into the lysis buffer for RNA extraction. Directly after sorting, samples were shock-frozen on dry ice. The sorted cells were kept at −80 °C before RNA isolation. A total of at least 20,000 monocytes per subset were sorted per group. RNA extraction was carried out using an RNAeasy mini kit (Qiagen, Hilden, Germany) and subsequently transformed to cDNA using superscript III (Invitrogen, Carlsbad, CA, USA); both steps were carried out according to the manufacturer’s instructions. These steps included DNase treatment and the verification of RNA quality and quantity.

### 4.5. QPCR Analysis of Sorted Monocytes

Following the manufacturer’s recommendations, the SYBR Green master mix (Roche Mannheim, Germany) was used for the quantitative real-time PCR of sorted monocyte subsets. Primers were selected from the Primer bank database (https://pga.mgh.harvard.edu/primerbank/, accessed on 10 April 2017). Real-time PCR was run on Applied Biosystems^®^ StepOne™ Real-Time PCR Systems (StepOne Software 2.0, Foster City, CA, USA) using the relative quantification of Ct values obtained from the threshold cycle number of a triplicate test; these values were then normalized relative to the healthy age-matched controls (N = 3). PPIA and RLP0 were used as housekeeping genes.

### 4.6. Human Muscle Biopsies

We analyzed muscle biopsies from 3 MD and Pred-MD patients. Dystrophin and sarcoglycan protein expression were assessed in muscle biopsies, confirming the diagnosis. Age-matched muscle specimens from healthy controls (N = 3) were taken from tissue designated as surgical waste during surgical procedures for congenital dislocation, with written informed consent provided. Biopsies are directly frozen in liquid nitrogen for immunobiological staining or qPCR-array analysis.

### 4.7. Hematoxylin and Eosin and Masson’s Trichrome Staining of Muscle Sections

Sections measuring 3 μM were cut from frozen human muscle biopsies embedded in plastic freezing molds (Cryomold, Torrance, CA, USA)containing O.C.T. (optimal cutting temperature compound) (Tissue-Tek, Torrance, CA, USA). Sections are stained with Hematoxylin and Eosin (HE) and with Masson’s trichrome (MT). Histology images were observed on an optical microscope, imaged at 10× and 20× magnification. To interpret the degree of inflammation and fibrosis in the samples, scoring was assigned by an anatomopathologist. A blinded evaluation of 3 different spots in the samples following Masson trichrome staining was carried out. The score for staining intensity was attributed as follows: 0% = absence of fibrosis; 20% = weak fibrosis; % > 40: highly abundant levels of fibrosis. The same was performed for HE staining, with a score varying from 0 for the absence of inflammation to a score of more than 3 for abundant levels of inflammatory cells.

### 4.8. RNA Extraction from Muscle Biopsies and Related cDNA Synthesis

Whole or dissociated portions of skeletal muscle tissue biopsy (10–20 mg) were removed from storage at 80 °C and placed in 1000 µL TRIzol homogenized with liquid nitrogen. RNA was then extracted from the samples. The total RNA concentration was quantified using nanodrop. RNA (0.5 ng/10 µL) was used to generate cDNA by reverse transcription using the RT2 profiler RT2 First Stand kit (Qiagen, Hilden, Germany), in accordance with the protocol provided. Muscle biopsies from healthy age-matched controls were obtained.

### 4.9. Analysis of Fibrosis-Related Genes by RT2 Profiler

The RT2 Human Fibrosis Profiler PCR Array, comprising 84 key genes encoding pro-fibrotic, anti-fibrotic, ECM remodeling, and inflammatory genes, was performed on the LightCycler 480 platform. The PCR array was quantified based on the CT value. A gene was considered undetectable when CT was >40. Two endogenous control genes, hypoxanthine phosphoribosyltransferase (HPRT1) and glyceraldehyde-3-phosphate dehydrogenase (GAPDH), were present in the PCR array, and they were used for normalization where the relative quantification of gene expression was applied.

### 4.10. PPI and Public Immunological Gene Expression Data

To explore the relationship between the key genes identified by the PCR array, we conducted PPI analysis using the Search Tool for the Retrieval of Interacting Genes/Proteins (STRING): https://string-db.org/, accessed on 5 July 2022. Only experimentally validated interactions with a combined score of >0.7 were set as significant. The ImmGen database was used to further explore the data array in single-cell data from different cell subset representations (https://www.immgen.org/Databrowser19/HumanExpressionData.html, accessed on 6 July 2022).

### 4.11. Statistical Analysis

Data are expressed as means ± SD. Normal distributions were analyzed using Shapiro–Wilk’s test, and subsequent comparisons were carried out by Student’s *t* or Mann–Whitney U tests, ANOVA, or Kruskal–Wallis tests with Tukey or Dunn corrections for multiple comparisons. The statistical significance level was set at 5% (*p* < 0.05). Data analysis was carried out with Prisma 9 software (Graphpad Inc., San Diego, CA, USA).

## Figures and Tables

**Figure 1 ijms-26-05992-f001:**
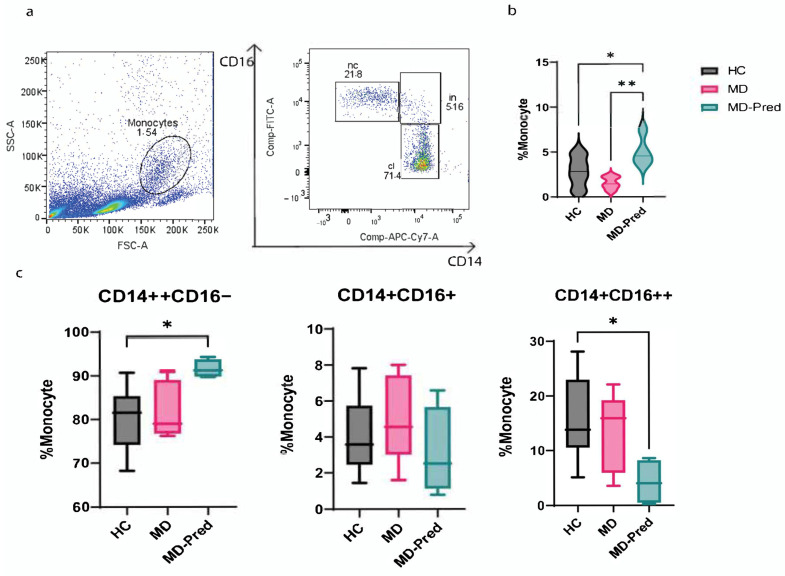
Analysis of total monocytes and subsets in healthy subjects (HCs), non-treated patients with muscle dystrophies (MD), and intermittent prednisone-treated MD patients (MD-Pred). (**a**) Flow cytometry gating strategy, (**b**) frequency of total monocytes, and (**c**) distribution of monocyte subsets. Statistical analysis was performed using one-way ANOVA. The *p*-value was set as * <0.05 and ** <0.01.

**Figure 2 ijms-26-05992-f002:**
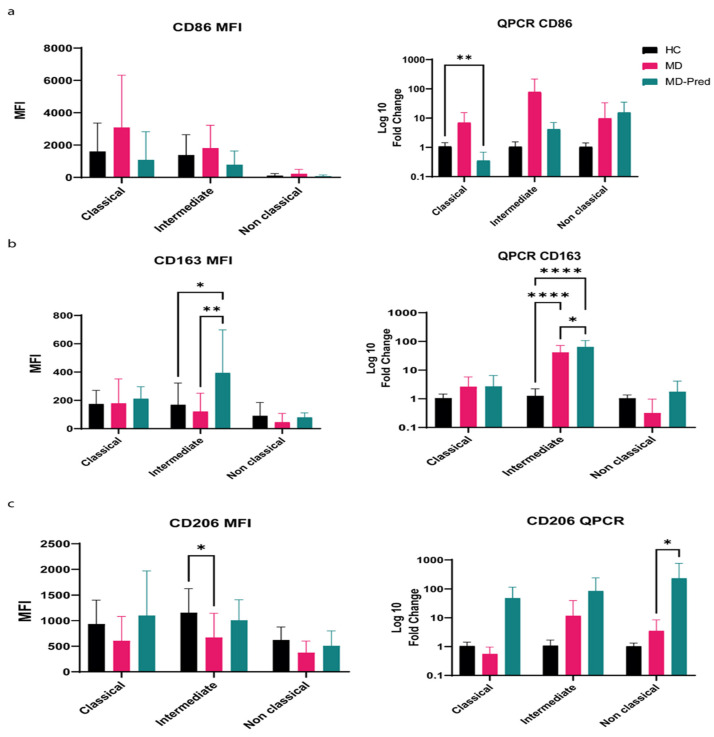
Expression of surface receptors CD86 (**a**), CD163 (**b**), and CD206 (**c**) following intermittent prednisone treatment in monocyte subsets in patients with muscular dystrophies and healthy controls, assessed by flow cytometry (**left panel**) and by QPCR (**right panel**). Two-way ANOVA with Tukey corrections for symmetric distributions and Kruskal–Wallis with Dunn corrections for asymmetric distributions were employed. MFI: Mean fluorescence intensity. * *p* < 0.05; ** *p* < 0.01; **** *p* < 0.0001.

**Figure 3 ijms-26-05992-f003:**
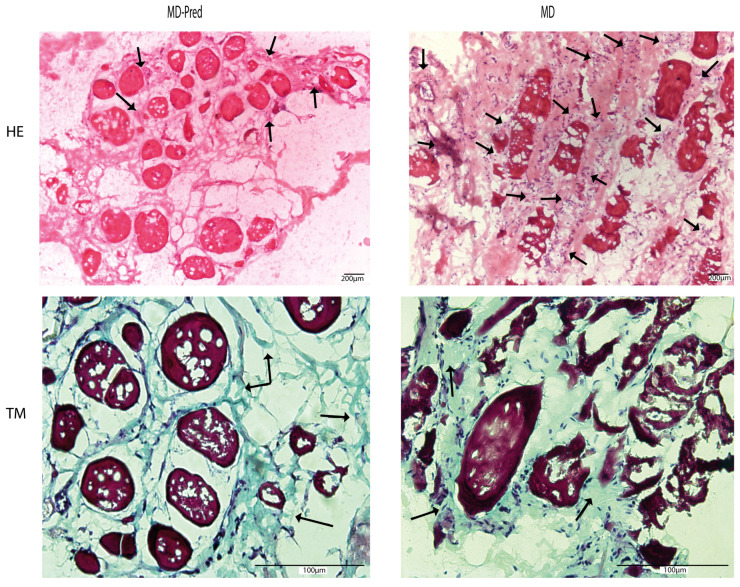
Hematoxylin–eosin (H&E) and Masson’s trichrome (M T)-stained representative biopsy sections from treated MD-Pred and non-treated groups of patients (N = 2 of each). Scale bar = 100 µm; arrows indicate potential sites of inflammation and fibrosis.

**Figure 4 ijms-26-05992-f004:**
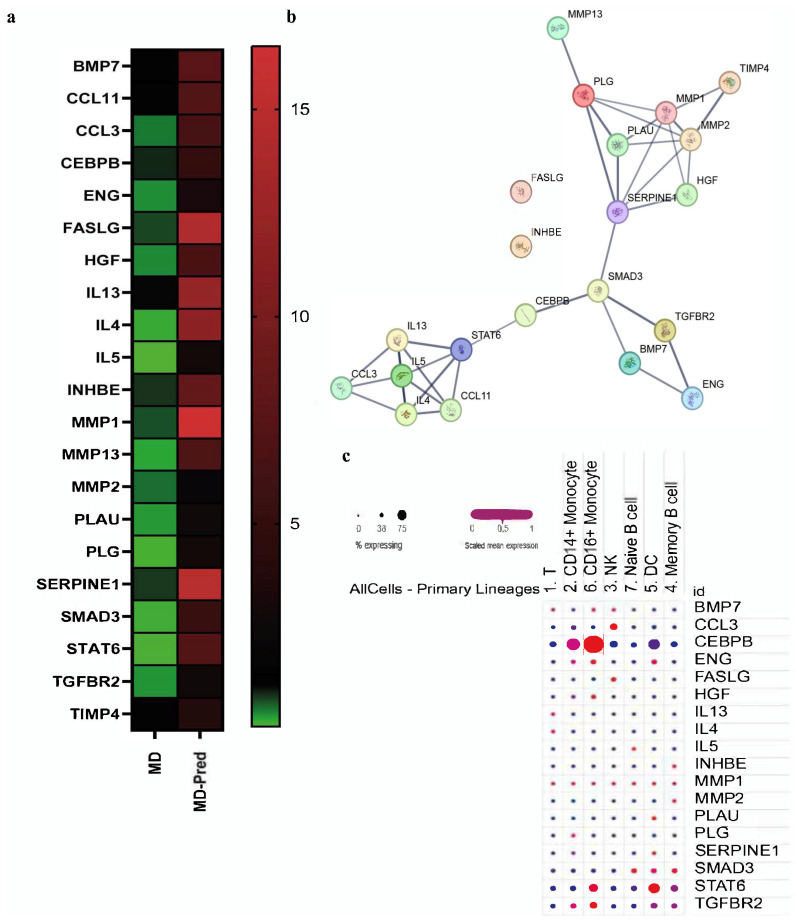
PCR-array and in silico analysis of fibrosis-related genes in MD and MD-Pred patients. (**a**) Heat maps of real-time RT2 Profiler fibrosis PCR array showing the normalized fold change in 21 genes, for which their differential expression was more than 2-fold between MD and MD-Pred groups. (**b**) STRING protein–protein interaction analysis of differentially expressed genes using a minimum required interaction score of 0.7 (confidence view). (**c**) Dot plot showing the expression of selected genes across different immune cell subtypes obtained from the Immgen database. The size of the dot encodes the percentage of cells expressing the gene, while the color intensity encodes the average expression level of “expressing” cells.

**Table 1 ijms-26-05992-t001:** Treated and non-treated prednisone patient description.

Groupe	ID	Age (y)	Age at Diagnosis (y)	Loss of Ambulation (Yes/No)	Genetic Anomalies
MD	DMD1	6	3	No	DMD deletion exon 45–50
	DMD6	10	6	No	DMD deletion exon 44
	DMD9	9	8	Yes	DMD deletion exon 8–48
	DMD11	7	6	No	DMD deletion exon 48–50
	DMD5	9	8	Yes	DMD deletion exon 3–11
	LGMD5	10	6	No	LGMD2C/R5 patient SGCG c.525delTC
	LGMD3EA	14	7	No	LGMD2C/R5 patient
	LGMD3EA2	14	7	No	LGMD2C/R5 patient
	LGMD7EA	12	10	Yes	LGMD2C/R5 patient SGCG c.525delTC
	LGMD7EA	8	6	No	LGMD2C/R5 patient SGCG c.525delTC
MD-Pred	DMD7	8	5	No	DMD deletion exon 45–54
	DMD10	11	9	Yes	DMD deletion exon 45–55
	DMD3	14	8	No	DMD deletion exon 45–47
	DMD8	9	4	No	DMD deletion exon 45
	LGMD2	11	2	Yes	LGMD2C/R5 patient SGCG c.525delTC
	LGMD6	11	9	Yes	LGMD2C/R5 patient SGCG c.525delTC

y: Year.

## Data Availability

All processed data are provided in the manuscript. The corresponding author can provide the raw data generated for this study upon reasonable request.

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
