# Peer review of "Profiles of Monocyte Subsets and Fibrosis-Related Genes in Patients with Muscular Dystrophy Undergoing Intermittent Prednisone Therapy"

_ijms, 2025, doi:10.3390/ijms26135992_

Round 1

Reviewer 1 Report

Comments and Suggestions for Authors

The work done by Asma Chikhaoui and cols. entitled “Intermittent Prednisone treatment shifts monocyte phenotypes and modifies the expression of fibrosis-related genes in patients with muscular dystrophies”, addresses a relevant and timely question regarding the immunomodulatory effects of intermittent prednisone treatment in muscular dystrophy (MD), focusing on peripheral monocyte phenotypes and fibrotic gene expression. Given the widespread use of glucocorticoids in MD and the ongoing debate around optimal dosing strategies, this work attempts to provide mechanistic insights into how intermittent prednisone modulates immune profiles and fibrosis. The study combines flow cytometry, gene expression analysis, and histological assessments, which, in theory, offers a multidimensional perspective. However, the manuscript suffers from several conceptual, methodological, and presentation weaknesses that limit its scientific impact and clarity. The conclusions are not fully supported by the data, and the small sample size makes it difficult to generalize findings.

Comments:

The introduction presents background information on MDs and prednisone use, the authors do not clearly justify why intermittent dosing specifically is of interest, nor do they reference contrasting effects seen in daily dosing regimens.

The dual focus on monocyte phenotypes and fibrosis-related gene expression is interesting but lacks integration.

The most critical limitation is the extremely small sample size: 9 patients total, subdivided into treated and untreated groups (n=4 and n=5, respectively), with only 3 muscle biopsies per group used for gene expression profiling. This raises serious concerns about statistical power and biological variability.

The claim that intermittent prednisone induces an M2-like shift is tenuous. The observed changes in CD163 and CD206 expression are modest and not consistently corroborated across cytometry and gene expression data. The discussion focuses on CD206 protein surface expression and mRNA levels without addressing the biological implications of this discrepancy.

The role of senescence is introduced late in the manuscript with reference to CEBPB, but without direct evidence (e.g., senescence-associated β-galactosidase activity or other senescence markers). Please explain

The authors claim that fibrosis is not significantly reduced in prednisone-treated patients, but this is based on qualitative staining from only three samples per group, with no quantification or blinded scoring. This weakens the conclusion substantially. Please clarify

The PCR array findings are potentially interesting but underpowered and over-interpreted. The use of pathway analysis (STRING, Reactome) is valuable, but the relevance of many of the reported genes (e.g., CEBPB) remains speculative without functional validation. Please support these findings with some previous publications.

The link between CEBPB expression and senescence or M2 polarization is suggested, but this interpretation is largely hypothetical. No functional assays were performed to validate CEBPB’s role in this context. Please clarify

Several p-values (e.g., P = 0.1098) are reported as significant when they are not. The authors should adhere to standard statistical thresholds and clearly report effect sizes and confidence intervals.

Author Response

1- The introduction presents background information on MDs and prednisone use, the authors do not clearly justify why intermittent dosing specifically is of interest, nor do they reference contrasting effects seen in daily dosing regimens

We would like to thank the reviewer for this insightful comment. We agree that the rationale for using intermittent prednisone treatment strategy needs to be more explained. Prednisone therapy has been used to treat Tunisian patients with LGMDR5 since the 1990s, which is not generally the mode of therapy for other types of LGMD. Moreover, this type of treatment is commonly used for Tunisian patients with DMD.

While some studies have reported that daily prednisone improves muscle strength and delays disease progression others have suggested that intermittent regimens are associated with several side-effects, but comparable functional benefits (Zulfiqar, E., et al.et al 2025). It becomes more evident that the choice of treatment protocol depends on the clinicians' prescription(Robert C. Griggs MD et al 2013), the patient's clinical status and the center in charge of these patients. In our case, intermittent administration with 10 days on/10 days off have been the unique prescribed regimen in the studied cohorts, in the hospital in which the study was conducted and it was independent of our choice.

Our study specifically investigates the biological particularities on the available samples from the patients who received intermittent prednisone treatment, or from patients who did not receive treatment  yet. We have revised the introduction (lines 74-81) to further clarify our study strategy and better introduce our issues in this study.

2- The dual focus on monocyte phenotypes and fibrosis-related gene expression is interesting but lacks integration.

Despite increasing evidence for the role of monocytes in fibrosis in murine models, there is still limited data obtained from human samples to support the role of monocytes in the pathogenesis of muscle dystrophy.

The role of monocytes during tissue regeneration or fibrosis has been well documented in studies in which we were involved in mouse models or in-vitro experiments (Rizzo, G., et al 2020). However, studies in association with these two components in patients with DMD and LGMDR5 are very limited and have not clearly addressed the issue. This we reported in the introduction (L93-99).

Data concerning the characterization of monocyte subsets in human samples, particularly in relation to fibrosis, remain limited if not absent. Our study aims to provide preliminary information on the involvement of monocytes in these two pathologies, and to report what is happening at the same time in relation to the dysregulation of fibrosis genes.

3- The most critical limitation is the extremely small sample size: 9 patients total, subdivided into treated and untreated groups (n=4 and n=5, respectively), with only 3 muscle biopsies per group used for gene expression profiling. This raises serious concerns about statistical power and biological variability.

We fully acknowledge the limitation posed by the small sample size and appreciate the reviewer’s emphasis on this critical point. However, this limitation reflects the inherent challenges associated with obtaining invasive muscle biopsies in pediatric patients with muscular dystrophy in an observational study. Importantly, all biopsies included in our study were obtained during corrective surgical procedures, rather than through procedures performed solely for research purposes. For ethical and the difficulty of convincing the parents to participate in the research project, have limited the availability of tissue samples.

Despite the limitations, we believe that these exploratory findings provide valuable insights and serve as a foundation for future, larger-scale investigations. We have added this to the discussion section in the revised manuscript (lines 369-386).

4- The claim that intermittent prednisone induces an M2-like shift is tenuous. The observed changes in CD163 and CD206 expression are modest and not consistently corroborated across cytometry and gene expression data. The discussion focuses on CD206 protein surface expression and mRNA levels without addressing the biological implications of this discrepancy

We agree that the modest and partially inconsistent changes in CD163 and CD206 expression across flow cytometry and transcriptomic data warrant careful interpretation. For these reasons, we report now that our findings suggest an evolution towards an M2-type macrophage phenotype, which would be keenly to be considered in the context of the pathophysiological mechanisms involved in these groups of muscular dystrophies.

Specifically, CD206 (mannose receptor) is known to undergo constitutive and rapid recycling between the plasma membrane and intracellular compartments. As early as 1980, Stahl and Schlesinger demonstrated that receptor-mediated endocytosis of mannose glycoconjugates involves continuous internalization and recycling of CD206, independently of new transcriptional activity. This increase in CD206 surface expression as detected by flow cytometry may reflect enhanced trafficking or membrane localization modulated by glucocorticoid rather than de novo synthesis

Similarly, the shedding into the soluble form  of both receptors can also occur without a parallel increase in mRNA due to the post transcriptional effect of systemic glucocorticoids, which stimulates receptor activity, as it have been stated before by Andriana Plevriti  et al 2024.

Taken together these data, we suggest that the uncoupling between mRNA and surface protein levels of CD206 and CD163 is not unexpected, particularly in the setting of glucocorticoid treatment. While we acknowledge that our findings do not conclusively prove an M2 polarization, they are rather consistent with a partial, perhaps transient, anti-inflammatory reprogramming of muscle-associated monocytes/macrophages.

To address this more clearly, we have revised the Discussion (L279-288)

5-The role of senescence is introduced late in the manuscript with reference to CEBPB, but without direct evidence (e.g., senescence-associated β-galactosidase activity or other senescence markers). Please explain

We thank the reviewer for this insightful observation. The inclusion of CEBPB as a senescence-associated transcription factor was based on its known role in orchestrating the senescence-associated secretory phenotype (SASP), particularly in tissue-resident immune and stromal cells during chronic inflammation and regeneration. Due to limited sample availability and resource constraints, we were unable to pursue further experimental validation in the current study. However, in silico exploration of ageing atlas data-base suggests that the expression of CEBPB was modestly increased (log 2 fc 0.42431) in aged human myotube in favor of the proposed hypothesis regarding is association to senescence. Additional related detail regarding aging atlas has been added to discussion section L 355-363

6- The authors claim that fibrosis is not significantly reduced in prednisone-treated patients, but this is based on qualitative staining from only three samples per group, with no quantification or blinded scoring. This weakens the conclusion substantially. Please clarify

We thank the reviewer for raising this important point. The assessment of fibrosis in our study was indeed based on Masson's trichrome staining. To minimize observer bias, fibrosis was evaluated through the assessment of three independent fields per section in a blinded scoring manner by an anatomic-pathologist and by the principle investigator of the study. The average score was used to support our observations.

We fully acknowledge that this approach is limited by the small sample size and lack of formal image quantification, which constrains the strength of the conclusion. Therefore, we have revised the manuscript to better reflect the preliminary nature of this finding and to avoid overinterpretation. The statement regarding the absence of significant fibrosis reduction has been rephrased to indicate that no clear pattern was observed under these specific experimental conditions, and we have added a sentence in the Discussion highlighting the need for more robust quantitative histological methods and larger sample cohorts to consolidate our results..(L293 , 300-302) as well as explained details regarding the quantification in the methods sections

7- The PCR array findings are potentially interesting but underpowered and over-interpreted. The use of pathway analysis (STRING, Reactome) is valuable, but the relevance of many of the reported genes (e.g., CEBPB) remains speculative without functional validation. Please support these findings with some previous publications

While we acknowledge that our PCR array findings are limited by sample size and currently lack direct functional validation, we believe that the pathway enrichment analysis using “STRING” and “Reactome” provides a valuable hypothesis-generating framework. Several genes we identified, such as CEBPB, although not functionally validated in our models, have established roles in related biological processes in the literature and more recently in a larger set of aging muscle data 2025, which conferring biological plausibility to our observations.

Although functional studies are necessary to definitively establish causality, prior studies give credence to the relevance of these targets such as CCAAT/enhancer-binding protein beta CEBPB (Zhang, Y., L. et al 2025). We have corrected and clarified that our preliminary work is   to highlight potentially relevant regulatory axes that warrant further experimental interrogation. L360-363

  1. Several p-values (e.g., P = 0.1098) are reported as significant when they are not. The authors should adhere to standard statistical thresholds and clearly report effect sizes and confidence intervals.

We thank the reviewer for pointing out these inaccuracies. We apologize for the typographical errors. All instances of incorrectly annotated significance markers (e.g., “*” or “**”) have been re-checked and aligned with corrected p-values.

Reviewer 2 Report

Comments and Suggestions for Authors

Major:

1.Limited Novelty

Insufficient clinical relevance: Prednisone has been widely used in the clinical treatment of muscular dystrophies (MDs), and its efficacy and side effects have been systematically studied (e.g., PMID: 35381069). The research topic of "the effect of intermittent prednisone treatment on monocyte phenotype polarization" has been extensively explored in previous literature (PMID: 19667757), which has clearly reported that glucocorticoids regulate the muscle fibrosis process by modulating the monocyte/macrophage phenotype. This study has not significantly broken through the existing cognitive framework.

Repetitive conclusion: Section 3.3 of the article mentions that "intermittent prednisone treatment did not significantly reduce muscle fibrosis in some patients",This finding has been confirmed by numerous previous studies (PMID: 19667757).

2.Conclusion validation is insufficient, and methodological flaws limit the research conclusion's reliability. From a clinical perspective, the sample size is undersized (n=9) and the design is non-randomized, potentially introducing selection bias. From a mechanistic perspective, the conclusion lacks experimental validation, as it hasn't been verified through in vitro experiments or animal models. The manuscript's clarity and coherence would improve by elaborating on the mechanism's importance and related opinions.

3.Statistical Methodology and Application

Questionable Statistical Significance:
In Figure 1b, the increase in total monocyte percentage (MD-Pred vs. MD group, P=0.1098)  is erroneously described as "significant". According to the figure legend (*P>0.05, **P<0.01), this P-value (>0.05) should be labeled as non-significant.

Inconsistent Figure Annotations:
In Figure 1c, the differences in classical monocytes (P=0.0147) and non-classical monocytes (P=0.0282) do not align with the figure legend’s definition (*P >0.05). Furthermore, the legend’s use of *P >0.05 and **P <0.01 contradicts standard conventions (*P <0.05, **P <0.01), leading to confusion.

4.Insufficient Mechanistic Explanation and Literature Support

IL4/IL13 Pathway Association:The claim that "the IL4/IL13 signaling pathway interacts with 8 out of 21 proteins (e.g., MMP2, IL13)" relies solely on Reactome database analysis (p=4.44E-16). However, experimental validation is absent, leaving the pathway’s role in MDs fibrosis unsubstantiated.

Minor:

1.Incomplete Data Interpretation

Discrepancy in Figure 4:Figure 4a lists 21 differentially expressed genes, but Figure 4c displays only 18 genes. The omission of three genes is neither annotated nor explained, raising concerns about data transparency.

2.Inadequate Discussion of Limitations

While the small sample size is briefly mentioned, its impact on result reliability (e.g., low statistical power and false-positive risks) is not thoroughly addressed. A dedicated discussion of these limitations is necessary.

3.Formatting Inconsistencies

Reference Style:Inconsistent citation formats are present (e.g., [27][28] in line 178 vs. [49][53] in line 234). Standardize all references to [27,28] or [49,53] throughout the text.

Missing Punctuation:Sentences in lines 309 and 330 lack terminal punctuation. A full-text review is required to correct such errors.

Author Response

  1. Limited Novelty

Insufficient clinical relevance: Prednisone has been widely used in the clinical treatment of muscular dystrophies (MDs), and its efficacy and side effects have been systematically studied (e.g., PMID: 35381069). The research topic of "the effect of intermittent prednisone treatment on monocyte phenotype polarization" has been extensively explored in previous literature (PMID: 19667757), which has clearly reported that glucocorticoids regulate the muscle fibrosis process by modulating the monocyte/macrophage phenotype. This study has not significantly broken through the existing cognitive framework.

Repetitive conclusion: Section 3.3 of the article mentions that "intermittent prednisone treatment did not significantly reduce muscle fibrosis in some patients",This finding has been confirmed by numerous previous studies (PMID: 19667757).

.

We appreciate the reviewer’s comment regarding the broader conceptual framework around glucocorticoid-mediated modulation of monocyte/macrophage phenotypes and their involvement in muscle fibrosis. Indeed, previous studies have established that glucocorticoids influence monocyte phenotyping, particularly in the context of tissue injury and fibrosis (Baoying Liu etl 2016) .  Our study contributes to this field in a way that is specific to the Maghreb population, particularly with regard to LGMDR5 patients who presented different manifestations to those seen in other muscular dystrophies that are not treated with prednisone as in the case of Europe (L66-70). Furthermore, to our knowledge, none of these studies have characterized the different monocyte subsets in muscular dystrophies (LGMDR5 and DMD). They have rather been studied in the context of investigating the effect of prednisone administration in other pathologies.

Conclusion validation is insufficient, and methodological flaws limit the research conclusion's reliability. From a clinical perspective, the sample size is undersized (n=9) and the design is non-randomized, potentially introducing selection bias. From a mechanistic perspective, the conclusion lacks experimental validation, as it hasn't been verified through in vitro experiments or animal models. The manuscript's clarity and coherence would improve by elaborating on the mechanism's importance and related opinions.

We agree that the small sample size and non-randomized design represent limitations, and we have now explicitly acknowledged this in the discussion section L369-386. We have therefore revised the abstract's conclusion, emphasizing the preliminary nature of the results and the need for larger randomized studies to confirm our findings

2- Statistical Methodology and Application

Questionable Statistical Significance:
In Figure 1b, the increase in total monocyte percentage (MD-Pred vs. MD group, P=0.1098)  is erroneously described as "significant". According to the figure legend (*P>0.05, **P<0.01), this P-value (>0.05) should be labeled as non-significant.

Inconsistent Figure Annotations:
In Figure 1c, the differences in classical monocytes (P=0.0147) and non-classical monocytes (P=0.0282) do not align with the figure legend’s definition (*P >0.05). Furthermore, the legend’s use of *P >0.05 and **P <0.01 contradicts standard conventions (*P <0.05, **P <0.01), leading to confusion.

We thank the reviewer for pointing out these inaccuracies. We apologize for the typographical errors. All instances of incorrectly annotated significance markers (e.g., “*” or “**”) have been re-checked and aligned with corrected p-values.

4.Insufficient Mechanistic Explanation and Literature Support

IL4/IL13 Pathway Association:The claim that "the IL4/IL13 signaling pathway interacts with 8 out of 21 proteins (e.g., MMP2, IL13)" relies solely on Reactome database analysis (p=4.44E-16). However, experimental validation is absent, leaving the pathway’s role in MDs fibrosis unsubstantiated

We thank the reviewer for pointing out the need for deeper mechanistic context and literature support. Due to limited sample availability and resource constraints, we were unable to pursue further experimental validation in the current study. Tin the section L321-335 we have added further clarifications from the literature that properly contextualize the results related to the IL4/IL13 signaling pathway and seamlessly communicate our preliminary findings  in muscular dystrophy.

.

Minor:

1.Incomplete Data Interpretation

Discrepancy in Figure 4:Figure 4a lists 21 differentially expressed genes, but Figure 4c displays only 18 genes. The omission of three genes is neither annotated nor explained, raising concerns about data transparency.

We would like to thank the reviewer to point that out. The missing genes did not have any available data in the consulted database

2.Inadequate Discussion of Limitations

While the small sample size is briefly mentioned, its impact on result reliability (e.g., low statistical power and false-positive risks) is not thoroughly addressed. A dedicated discussion of these limitations is necessary.

Further clarifications regarding the limitation of this study has been added to the discussion L369-386

3.Formatting Inconsistencies

Reference Style:Inconsistent citation formats are present (e.g., [27][28] in line 178 vs. [49][53] in line 234). Standardize all references to [27,28] or [49,53] throughout the text.

Missing Punctuation:Sentences in lines 309 and 330 lack terminal punctuation. A full-text review is required to correct such errors.

Typos has been corrected accordingly.

Round 2

Reviewer 1 Report

Comments and Suggestions for Authors

Although most of the comments and criticism have been addressed. The manuscript is lacking of a deeper clarity mainly in the introduction and discussion sections. Please, when possible and applicable, integrate the responses to the criticism to the full manuscript.

Author Response

1-Although most of the comments and criticism have been addressed. The manuscript is lacking of a deeper clarity mainly in the introduction and discussion sections. Please, when possible and applicable, integrate the responses to the criticism to the full manuscript.

We thank the reviewer for the valuable feedback. We have revised both the introduction and discussion sections to include additional information, to provide more clarity, and to respond to criticisms (L74-76) (L299-303)(381-385)

Reviewer 2 Report

Comments and Suggestions for Authors

Major:

1.Lack of animal andin vitro cell experiments. It is recommended that the authors incorporate additional methods to strengthen the validation of their points and address the small sample size issue。

Minor:

1.Incorrect punctuation and extra highlighting are present in the manuscript. The authors are advised to carefully review and revise the entire text (e.g. line 203-205)

Comments on the Quality of English Language

There are tense inconsistencies and grammatical errors (e.g. in L257-260). Please carefully review and correct the grammar throughout the manuscript.

Author Response

1.Lack of animal and in vitro cell experiments. It is recommended that the authors incorporate additional methods to strengthen the validation of their points and address the small sample size issue

We fully acknowledge the importance of incorporating in vitro and animal experiments to strengthen mechanistic insights and address limitations due to the small sample size. We have previously used the mdx mouse, which mimics DMD, to study fibrosis (PMID: 30463013 ). However, despite its wide use, this model does not fully recapitulate the pathophysiology of DMD and may not reflect the specific effects of intermittent glucocorticoid regimens used in clinical practice in patients. In addition, differences in disease progression  and genetics between species limit direct extrapolation of results.

Furthermore, due to  restricted access to fresh samples to establish cell cultures and in vitro experimentations we were unable to conduct further experiments at this stage. Our study is based on rare biological material  limited by ethical procedures, hence the small number of samples.

The results obtained in this study are preliminary but valuable as they are obtained in a human North African population using a treatment protocol different from that used in other countries .( L74-76 L80-81)

Minor:

1.Incorrect punctuation and extra highlighting are present in the manuscript. The authors are advised to carefully review and revise the entire text (e.g. line 203-205)

We thank the reviewer for pointing this out. We have carefully reviewed the manuscript for punctuation issues, formatting inconsistencies, and unintentional text highlighting throughout the document.

2- Comments on the Quality of English Language

There are tense inconsistencies and grammatical errors (e.g. in L257-260). Please carefully review and correct the grammar throughout the manuscript.

We thank the reviewer for this helpful observation. We have carefully reviewed the manuscript and corrected grammatical issues